# INSTRUCTPRO: NATURAL LANGUAGE GUIDED LIGAND-BINDING PROTEIN DESIGN

## ABSTRACT

Designing ligand-binding proteins with precise functions is fundamental to advances in biology and chemistry, yet existing AI approaches are limited by scarce protein–ligand complex data. Meanwhile, abundant text descriptions of protein–ligand interactions remain underutilized. We introduce InstructPro, a family of generative models that design proteins from natural language instructions and ligand formulas. InstructPro produces protein sequences consistent with specified functional descriptions and ligand targets. To enable training and evaluation, we develop InstructProBench, a large-scale dataset of 9.6 million (function description, ligand, protein) triples. We train two model variants — InstructPro-1B and InstructPro-3B — that substantially outperform strong baselines. InstructPro-1B achieves design success rates of 2.46% (seen ligands) and 3.14% (zero-shot), while InstructPro-3B reaches 5.06% and 3.93%, respectively. These results demonstrate the potential of natural language–guided generative modeling to expand protein design capabilities beyond traditional data limitations.

## 1 INTRODUCTION

Proteins are fundamental to all forms of life and execute a wide range of biological functions. Many of these functions are mediated through specific protein–ligand interactions, such as catalyzing chemical reactions (Dauparas et al., 2025), sensing small molecules that regulate signal transduction pathways (Wang et al., 2020), or recognizing cell-surface receptors to enable targeted drug delivery to diseased cells (Srinivasarao & Low, 2017). The ability to custom design ligand-binding proteins thus holds the promise of expanding protein functionality and enabling a wide range of applications, including therapeutics, diagnostics, industrial enzymes, biosensors, and molecular tools for synthetic and chemical biology (Tinberg et al., 2013; Bunzel et al., 2021; Rottinghaus et al., 2022; Pennington et al., 2023).

Recent advances in deep learning have led to remarkable progress in predicting and designing protein–ligand interactions (Ahmed et al., 2021; Chatterjee et al., 2023; Dauparas et al., 2025; Rezaei et al., 2020; Rube et al., 2022; Zhao et al., 2020). These methods have enabled the direct design of proteins tailored to specific ligands, reducing reliance on high-throughput experimental screening. However, despite these successes, current approaches remain limited by the relatively small number of publicly available protein–ligand complex structures (approximately 20,000), leaving substantial room for further exploration in this area (Chakraborti et al., 2021; Siebenmorgen et al., 2024; Cao et al., 2025). Encouragingly, while structural data are scarce, ligand identities are annotated for many proteins in large-scale databases such as UniProtKB (uni, 2025). Moreover, a wealth of human-curated textual knowledge exists, describing protein functions and ligand-binding properties in detail. Together, these non-structural resources offer an underutilized opportunity to support large-scale ligand-binding protein design.

To address this challenge, we introduce InstructPro, a natural language-instructable (e.g., Please design a protein which catalyzes phosphoribosyl pyrophosphate to diphosphate.), multimodal framework for the design of ligand-binding proteins. InstructPro comprises four key components: a text encoder, a ligand encoder, a shared memory module and a protein decoder. The text encoder encodes the combined human instruction and protein function description into comprehensive semantic representations, based on which a shared memory module extracts critical protein function information to facilitate more targeted and efficient protein design. The ligand encoder processes

ligand representations in SMILES format. Conditioned on both the extracted textual semantics and ligand representations, the protein decoder generates a protein sequence that aligns with the specified function description and is capable of binding the target ligand. This architecture enables InstructPro to design ligand-binding proteins directly from textual instructions, without requiring structural input.

To summarize, our contributions are listed as follows:

- We present InstructPro, the natural language instruction-following model and training approach for designing ligand-binding proteins directly from function descriptions and ligand formulas.
- We curate InstructProBench, a large-scale dataset containing 9,592,829 (function description, ligand, protein) triples, enabling scalable training and evaluation for instruction-guided protein design.
- We develop two model variants — InstructPro-1B (1 billion parameters) and InstructPro-3B (3 billion parameters) — that consistently outperform strong baselines. InstructPro-1B achieves the highest design success rates of 2.46% on seen ligands and 3.14% in zero-shot settings, while InstructPro-3B further improves to 5.06% and 3.93%. These results demonstrate the effectiveness of natural language guidance in generating functional ligand-binding proteins.

The code is available at `https://anonymous.4open.science/r/InstructPro-9B05`. Designed proteins are provided in the supplementary materials.

## 2 RELATED WORK

**Ligand-Binding Protein Design.** Designing proteins that bind to specific ligands requires known information about the key interacting residues. Therefore, most work in this area has focused on redesigning proteins based on known structural motif information (Hellinga & Richards, 1991; Tinberg et al., 2013; Dou et al., 2017; Nguyen et al., 2024). However, recent advances in *de novo* protein design have enabled the generation of proteins tailored to specific ligands (Krishna et al., 2024; Lu et al., 2024; Zambaldi et al., 2024; Ahern et al., 2025). For instance, generative models such as RFDiffusionAA (Krishna et al., 2024) can produce novel protein backbones tailored to bind a specific ligand, often by building around a defined ligand pose. Once such candidate proteins are designed, they can be further screened by molecular dynamics simulations (Barros et al., 2019), docking (Corso et al., 2024) or by using co-folding methods (Krishna et al., 2024; Abramson et al., 2024). Despite the advancements in such *de novo* design methods, precisely controlling for other specific functional attributes of the protein, beyond just binding the ligand, remains a significant area for development (Chu et al., 2024).

**Natural Language Guided Protein Design.** Recently, there has been a shift from controlled protein design using methods such as keyword tags (Nijkamp et al., 2023; Hayes et al., 2025), learned prefix tokens (Luo et al., 2023) or task specific instructions (Lv et al., 2025) to make up much more general language-guided protein design methods (Guo et al., 2024; Dai et al., 2024; Praljak et al., 2024; Riley et al., 2025; Xia et al., 2025; Liu et al., 2025). For instance, BioM3 (Praljak et al., 2024) generates protein sequences through an auto-regressive diffusion model. Dai et al. (2024) proposes Pinal, a model that generates three-dimensional (3D) structure tokens first and then generates protein sequences. They find that such two-stage approach helps to limit the search space, rather than end-to-end text-to-sequence generation. Furthermore, Riley et al. (2025) proposes a transformer-based model (MP4) for end-to-end text-to-sequence generation, and have performed experimental validation to test the ability of the generated protein sequences to stably express. Although these methods demonstrate the capability of natural language-guided protein design, they do not incorporate conditioning on small molecules, which is necessary, for example, in designing ligand-binding proteins.

Accounting for the limitations in both of these use cases, we propose InstructPro, a language-guided ligand-binding protein design model, that goes beyond understanding just protein sequences.

## 3 PROPOSED METHOD: INSTRUCTPRO

### 3.1 PROBLEM DEFINITION

A protein is composed of a linear chain of amino acids, covalently linked by peptide bonds and folded into a specific 3D structure. Let $\mathcal{A}$ denote the set of the 20 standard amino acids. We represent a

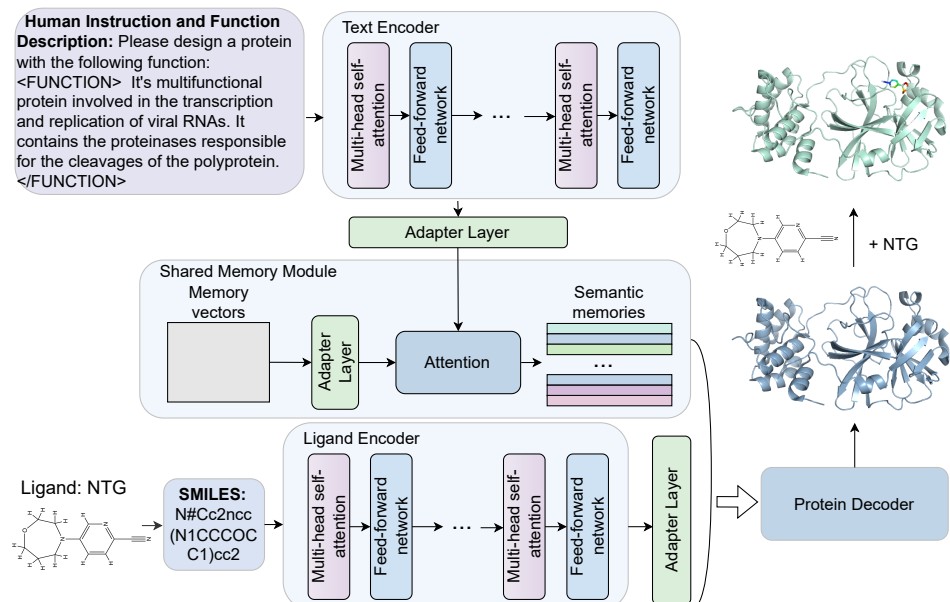

Figure 1: The overall architecture of InstructPro. The text encoder processes human-provided instructions and protein function descriptions in natural language. The shared memory module extracts contextual semantics according to function descriptions. The ligand encoder encodes molecular representations from the given ligand SMILES strings, capturing the chemical context of the target ligand. Conditioned on both the critical contextual semantics and ligand representations, the protein decoder generates a protein sequence that aligns with the functional specification and is able to bind to the target ligand.

protein sequence of length $N_r$ as $\boldsymbol{r} = \{r_1, r_2, \ldots, r_{N_r}\} \in \mathcal{A}^{N_r}$, where each residue $r_i \in \mathcal{A}$ for $i \in \{1, \ldots, N_r\}$. To guide the design, we concatenate a user-provided instruction and a corresponding protein function description into a natural language sequence $\boldsymbol{w} = \{w_1, w_2, \ldots, w_{N_w}\}$ of length $N_w$, where the function description is marked with special tokens <FUNCTION> and </FUNCTION>. $w_j \in \mathcal{V}$ for $j \in \{1, 2, ..., N_w\}$ and $\mathcal{V}$ denotes the natural language token vocabulary. Each ligand is represented by a SMILES string $\boldsymbol{s} = \{s_1, s_2, \ldots, s_{N_s}\}$ with length $N_s$, and each token $s_k \in \mathcal{S}$ for $k \in \{1, 2, ..., N_s\}$, with $\mathcal{S}$ denoting the SMILES token alphabet.

The task is formulated as follows: given a user-provided instruction and protein function description $\boldsymbol{w}$, and a target ligand $\boldsymbol{s}$, generate a protein sequence $\boldsymbol{r}$ that binds to the ligand and satisfies the described function. Formally, we aim to learn a conditional generative model $P(\boldsymbol{r} \mid \boldsymbol{w}, \boldsymbol{s}; \theta)$, parameterized by $\theta$.

## 3.2 OVERALL MODEL ARCHITECTURE

The overall architecture of InstructPro comprises four core components: a text encoder, a ligand encoder, a shared memory module, and a protein decoder, as illustrated in Figure 1. The text encoder processes natural language input, including human instructions and protein function descriptions, to produce a sequence embeddings. The shared memory module uses a fixed set of memory eliciting vectors to query text embeddings and produce semantic memory embeddings. The ligand encoder captures the chemical context of the target ligand from its SMILES strings. Conditioned on both the semantic memory and ligand representations, the protein decoder generates a protein sequence that satisfies the specified functional and ligand-binding requirements. In the following subsections, we provide detailed introductions of each module, followed by an explanation of the training objective.

### 3.3 TEXT ENCODER AND SHARED MEMORY MODULE

The text encoder module encodes the combined human instruction and protein function description into comprehensive semantic representations. It consists of a $N_t$-layer Transformer encoder similar to BERT model (Devlin et al., 2018).

Each layer in the text encoder comprises a multi-head self-attention (MHA) sub-layer and a feed-forward network (FFN), both followed by residual connections and layer normalization. The representation of each token in the input text is computed as:

$$
\begin{aligned}
\boldsymbol{h}_{t,i}^0 &= \boldsymbol{E}_t \cdot \text{onehot}(w_i) \\
\tilde{\boldsymbol{h}}_{t,i}^{l+1} &= \text{LayerNorm}\left(\text{MHA}(\boldsymbol{h}_{t,i}^l, \boldsymbol{H}_t^l) + \boldsymbol{h}_{t,i}^l; \theta_t\right) \\
\boldsymbol{h}_{t,i}^{l+1} &= \text{LayerNorm}\left(\text{FFN}(\tilde{\boldsymbol{h}}_{t,i}^{l+1}) + \tilde{\boldsymbol{h}}_{t,i}^{l+1}; \theta_t\right)
\end{aligned}
\tag{1}
$$

where $\boldsymbol{h}_{t,i}^l$ is the $i^{th}$ token input representation at the $l^{\text{th}}$ layer and $\boldsymbol{h}_{t,i}^0$ is the $i$-th token embedding in the text sequence. $\boldsymbol{H}_t^l = [\boldsymbol{h}_{t,1}^l, \boldsymbol{h}_{t,2}^l, ..., \boldsymbol{h}_{t,N_w}^l]^T$ denotes the full sequence input at layer $l$. The embedding matrix $\boldsymbol{E}_t$ maps each input token $w_i$ into a dense vector of dimensionality $D_t$. All parameters in the text encoder are denoted by $\theta_t$.

To mitigate the inefficiency introduced by long textual function descriptions which may span thousands of tokens, we introduce a shared memory module to elicit key semantic information. Specifically, we introduce a set of $N_m$ (trainable) memory eliciting vectors $\boldsymbol{M}$, which serve as queries in an attention mechanism to extract essential textual features from the final layer output embeddings of the above text encoder:

$$
\hat{\boldsymbol{H}}_t = \text{Softmax}(W_q \boldsymbol{M} \cdot W_k \boldsymbol{H}_t^{L_t}) W_v \boldsymbol{H}_t^{L_t}
\tag{2}
$$

where $W_q$, $W_k$, and $W_v$ are learnable projection matrices with dimensionality $D_t \times D_t$. This mechanism enables the model to selectively retain and compress critical semantic content from the input instructions and function descriptions, facilitating more targeted and efficient protein sequence generation.

### 3.4 LIGAND ENCODER

To model the binding-ligand information, we employ another Transformer encoder with $N_g$ layers to encode the ligand's SMILES strings. The resulting contextual representations capture structural and chemical properties of the ligand, providing essential cues to guide the generation of protein sequences capable of binding the desired target.

Specifically, the encoding of each token in the SMILES string is computed as:

$$
\begin{aligned}
\boldsymbol{h}_{g,i}^0 &= \boldsymbol{E}_g \cdot \text{onehot}(s_i) \\
\tilde{\boldsymbol{h}}_{g,i}^{l+1} &= \text{LayerNorm}\left(\text{MHA}(\boldsymbol{h}_{g,i}^l, \boldsymbol{H}_g^l) + \boldsymbol{h}_{g,i}^l; \theta_g\right) \\
\boldsymbol{h}_{g,i}^{l+1} &= \text{LayerNorm}\left(\text{FFN}(\tilde{\boldsymbol{h}}_{g,i}^{l+1}) + \tilde{\boldsymbol{h}}_{g,i}^{l+1}; \theta_g\right),
\end{aligned}
\tag{3}
$$

where $\boldsymbol{h}_{g,i}^l$ denotes the representation of the $i^{th}$ SMILES token at the $l^{\text{th}}$ layer and $\boldsymbol{h}_{g,i}^0$ is the $i$-th SMILES token embedding. $\boldsymbol{H}_g^l = [\boldsymbol{h}_{g,1}^l, \boldsymbol{h}_{g,2}^l, ..., \boldsymbol{h}_{g,N}^l]^T$ represents the full sequence input at layer $l$. The embedding matrix $\boldsymbol{E}_g$ maps one-hot encoded token $s_i$ to continuous vectors of dimensionality $D_g$. $\theta_g$ denotes the trainable parameters in the ligand encoder.

The ligand representations from the final layer are subsequently integrated with the semantic memory vectors, serving as conditioning inputs to the protein decoder. This fusion ensures that the generated protein sequence reflects both the functional intent and the structural requirements necessary for ligand binding.

### 3.5 PROTEIN DECODER

The protein decoder is designed to generate a protein sequence that both aligns with the given protein function description and exhibits binding ability to the specified ligand. Specifically, we adopt a Transformer decoder-based architecture (Vaswani et al., 2017) to autoregressively generate the protein

Table 1: Hyper-parameter settings of the InstructPro family. For the text encoder, $N_t$, $\text{Head}_t$, and $D_t$ denote the number of layers, attention heads, and embedding dimension, respectively. For the ligand encoder, $N_g$, $\text{Head}_g$, and $D_g$ represent the same parameters. For the protein decoder, $N_p$, $\text{Head}_p$, and $D_p$ follow the same notation. $N_m$ denotes the memory eliciting vector size.

| Models | Parameters | $N_t$ | $\text{Head}_t$ | $D_t$ | $N_m$ | $N_g$ | $\text{Head}_g$ | $D_g$ | $N_p$ | $\text{Head}_p$ | $D_p$ |
|---|---|---|---|---|---|---|---|---|---|---|---|
| InstructPro-1B | 964M | 12 | 12 | 768 | 64 | 6 | 12 | 768 | 27 | 16 | 1,536 |
| InstructPro-3B | 2.98B | 12 | 12 | 768 | 64 | 6 | 12 | 768 | 32 | 32 | 2,560 |

sequence. At every decoding step, prefix tuning (Li & Liang, 2021) is applied to get the conditional probability distribution of each residue $r_i$. The overall model contains $N_p$ layers of unidirectional Transformer layer, with embedding size $D_p$:

$$\hat{\boldsymbol{H}}_g = \text{FFN}(\boldsymbol{H}_g^{N_g}), \ \boldsymbol{h}_{p,i} = F_{\text{protein}}(\boldsymbol{r}_{<i}|\hat{\boldsymbol{H}}_t, \hat{\boldsymbol{H}}_g; \theta_p), \ P(r_i) = \text{Softmax}(\boldsymbol{h}_{p,i}) \quad (4)$$

where $F_{\text{protein}}$ denotes the protein decoder network parameterized by $\theta_p$, and $\boldsymbol{r}_{<i} = \{r_1, r_2, ..., r_{i-1}\}$ is the partial sequence generated before position $i$. The vectors $\hat{\boldsymbol{H}}_t$ and $\hat{\boldsymbol{H}}_g$ represent the semantic memory vectors of the text and the final chemical contexts from the ligand inputs, respectively. The adapter module FFN projects the ligand features from the ligand encoder into the residue representation space, ensuring modality alignment.

The model is trained to minimize the negative log-likelihood of the target protein sequence:

$$\mathcal{L} = \sum_{i=1}^{N_r} -\log P(r_i) \quad (5)$$

where $N_r$ denotes the length of the designed protein sequence. The prefix tuning strategy encourages the protein decoder to generate protein sequences that are both functionally consistent with the textual description and structurally complementary with the target ligand.

## 3.6 MODEL IMPLEMENTATION

We implement and train two model variants: InstructPro-1B and InstructPro-3B. Both models use a $N_t = 12$ layer text encoder with 12 attention heads and an embedding dimensionality $D_t = 768$. Its parameters are initialized with the pretrained PubMedBERT model (Gu et al., 2020). Both models use a $N_g = 6$ layer ligand encoder with 12 attention heads and an embedding dimensionality $D_g = 768$. Its parameters are initialized with the pretrained SMILES-based RoBERTa model (Ahmad et al., 2022). We use $N_m = 64$ memory eliciting vectors in the shared memory module, and a single layer feed-forward network in the adapter. InstructPro-1B uses $N_p = 27$ autoregressive Transformer decoding layers with 16 attention heads, and embedding dimensionality $D_p = 1,536$. InstructPro-3B uses $N_p = 32$ and $D_p = 2,560$. The weights of both variants are initialized from ProGen2-base and ProGen2-BFD90, respectively (Nijkamp et al., 2023). The total number of parameters is approximately 964 million for InstructPro-1B and 2.98 billion for InstructPro-3B. During training, the parameters of the text encoders and ligand encoders are frozen, and all other model weights are updated. Table 1 summarizes the hyper-parameter settings of the InstructPro family.

Both InstructPro-1B and InstructPro-3B are trained for 800,000 steps using eight NVIDIA H100 GPUs. We adopt the Adam optimizer (Kingma & Ba, 2014) with a linear learning rate warm-up over the first 10,000 steps, followed by linear decay. The learning rate is set to 1e-3. The batch size is set to 16,384 tokens for InstructPro-1B and 6,144 tokens for InstructPro-3B, respectively. During inference, we employ greedy decoding to produce the output sequence. The maximum sequence length is set to 1,024 residues.

## 3.7 INSTRUCTPROBENCH CONSTRUCTION

To train and evaluate two variants of InstructPro, we construct InstructProBench, a large-scale dataset of protein–ligand–function triples. We begin by extracting all natural language function annotations from UniProt (uni, 2025) and retain only those entries with recorded binding ligands. This results in 9,592,829 triples containing both functional descriptions and associated ligands, corresponding to 7,535,394 unique UniProt entries (as some proteins are associated with multiple ligands).

Table 3: Performance on seen ligands. For limited space, RF2 denotes RFdiffusion2 and AF3 denotes AlphaFold3. **The results demonstrate that InstructPro-1B achieves higher success rate than baseline methods, and scaling the model to 3B yields additional performance gains.**

| Models | RMSD (↓) | % < 2Å RMSD (↑) | DockSR (↑) | ipTM (↑) | PAE (↓) | pLDDT (↑) | SR (↑) |
|---|---|---|---|---|---|---|---|
| ProGen2 (764M) | 7.773 | 56.40% | 2.06% | 0.589 | 10.496 | 71.655 | 0.26% |
| ESM3 (1.4B) | 8.872 | 57.06% | 2.66% | 0.624 | 10.181 | 75.368 | 0.00% |
| Pinal (1.2B) | 7.147 | 62.53% | 1.80% | 0.751 | 6.588 | 81.419 | 0.06% |
| RF2+LigandMPNN | 10.354 | 8.13% | 2.86% | 0.693 | 13.356 | 66.501 | 0.00% |
| AF3+LigandMPNN | **3.068** | 75.20% | 2.93% | 0.856 | 2.371 | 88.200 | 2.33% |
| InstructPro-1B | 3.865 | **81.00%** | **3.13%** | 0.918 | 1.893 | 89.601 | 2.46% |
| InstructPro-3B | **2.763** | **85.06%** | **7.86%** | 0.877 | 2.261 | 87.586 | **5.06%** |

Table 4: Performance on unseen ligands (zero-shot setting). For limited space, RF2 denotes RFdiffusion2 and AF3 denotes AlphaFold3. **InstructPro-1B consistently outperforms all baseline methods on every metric. Moreover, scaling the model to 3B yields further performance improvements.**

| Models | RMSD (↓) | % < 2Å RMSD (↑) | DockSR (↑) | ipTM (↑) | PAE (↓) | pLDDT (↑) | SR (↑) |
|---|---|---|---|---|---|---|---|
| ProGen2 (764M) | 7.286 | 51.96% | 2.36% | 0.686 | 9.079 | 69.176 | 2.36% |
| ESM3 (1.4B) | 5.249 | 57.48% | 7.87% | 0.774 | 5.757 | 86.857 | 3.14% |
| Pinal (1.2B) | 4.804 | 60.62% | 0.00% | 0.814 | 3.247 | 87.157 | 0.00% |
| RF2+LigandMPNN | 9.928 | 10.41% | 4.16% | 0.745 | 9.898 | 75.491 | 0.00% |
| AF3+LigandMPNN | 8.360 | 34.64% | 3.93% | 0.868 | 2.923 | 89.862 | 0.00% |
| InstructPro-1B | **4.143** | **60.62%** | 3.93% | 0.869 | 2.638 | 90.518 | 3.14% |
| InstructPro-3B | **2.998** | **62.20%** | 3.93% | 0.882 | 2.439 | 90.207 | **3.93%** |

To group proteins with similar sequences, we perform clustering using MMseqs2 (Steinegger & Söding, 2017) at a 30% sequence identity threshold, yielding 22,594 clusters. To ensure diversity in the validation and test sets, we stratify the clusters by size (number of sequences per cluster) and allocate: 20 clusters from the 1–500 sequence count range, 10 from 501–1000, 5 from 1001–2500, and 1 from clusters containing over 2500 sequences to each of the validation and test sets. The remaining clusters are

Table 2: InstructProBench size.

| Split | Size |
|---|---|
| Training | 9,586,972 |
| Validation | 3,263 |
| Test (Seen Ligand) | 1,500 |
| Test (Unseen Ligand) | 127 |

used for training. This stratification ensures that the evaluation sets contain a mix of popular (large) and rare (small) protein sequence clusters. We further categorize the resulting examples based on whether their ligands are included in the training set. For validation, a set comprising 3,263 examples is constructed, of which the binding ligands do not present in the training set. For testing, we split 1,500 cases with seen ligands and 127 cases with ligands that are not observed in training set. The dataset size is summarized in Table 2 and detailed dataset statistics are provided in Appendix Table 5.

## 4 EXPERIMENTS

### 4.1 BASELINE MODELS

We benchmark InstructPro against two categories of protein design approaches: natural language–guided models and structure-based ligand-binding models. Natural language–guided baselines include: (1) **Pinal** (Dai et al., 2024), an encoder–decoder framework that generates protein sequences conditioned on natural language descriptions of the intended function. (2) **ESM3** (Hayes et al., 2025), a multimodal generative model capable of conditioning on functional annotations expressed in natural language. Structure-based ligand-binding baselines include: (3) **RFdiffusion2** (Ahern et al., 2025)+**LigandMPNN** (Dauparas et al., 2025), where RFdiffusion2 first designs a protein backbone structure given the ligand structure in the ground-truth complex, which is then used by LigandMPNN to generate the corresponding protein sequence. (4) **AlphaFold3** (Abramson et al., 2024)+**LigandMPNN**, which first predicts the structure of the ground-truth protein–ligand complex and then applies LigandMPNN to generate a sequence consistent with the folded structure. Finally,

since our protein design is initialized with ProGen2, we also compare against (5) **ProGen2-base**. To ensure a fair comparison, we generate protein candidates for all baseline models using their official codes. More baseline implementation details are provided in Appendix C.1.

## 4.2 EVALUATION METRICS

To evaluate whether the designed proteins achieve the targeted function while maintaining structural stability, we assess them along two dimensions: functional performance and structural reliability. Functional performance is conducted using the following metrics: (1) **RMSD** between the designed protein and the ground-truth structure, as structural similarity is generally indicative of preserved function. (2) **% < 2Å RMSD**, the proportion of designed proteins with RMSD below 2Å. (3) **docking success rate (DockSR)** defined as the fraction of designed proteins achieving high-confidence docking (DiffDock score > 0) using DiffDock (Corso et al., 2024). (4) **AlphaFold3 ipTM** which measures interface interactions and reflects binding affinity. Structural reliability is assessed using: (5) **AlphaFold3 PAE** which evaluates the reliability of predicted structures, with lower values indicating greater folding reliability. (6) **AlphaFold3 pLDDT** where scores above 80 suggest stable structures. Finally, we compute the (6) **overall success rate (SR)** as the proportion of designs meeting all criteria: RMSD < 2Å, DiffDock score > 0, ipTM $\geq$ 0.8, PAE $\leq$ 10, and pLDDT $\geq$ 80.

## 4.3 MAIN RESULTS

The performance of the model on both seen and unseen ligands is summarized in Table 3 and Table 4, respectively. **InstructPro-1B achieves the highest success rate among all baseline methods in both settings, with scores of 2.46% and 3.14%.** For functional evaluation, InstructPro-1B not only attains a higher proportion of designed proteins with RMSD < 2Å compared to the ground truth, but also achieves superior docking success rates and AlphaFold3 ipTM scores with the target ligand. These results confirm that InstructPro is capable of designing proteins that both exhibit the intended function and bind to the specified ligand. In addition, proteins designed by InstructPro-1B achieve structure reliability scores that surpass widely accepted thresholds, such as PAE < 10 (Bennett et al., 2023) and pLDDT > 80 (Guo et al., 2022), demonstrating the ability of InstructPro to generate proteins with stable structures. We observe consistent trends in the zero-shot setting, highlighting the model's generalizability and its capability to design proteins that bind to unseen ligands during training.

We find that the RFdiffusion2+LigandMPNN pipeline yields higher RMSD values than other methods, likely because RFdiffusion2 generates novel structures that, despite higher RMSD, remain functionally relevant. To test this, we report success rates without the RMSD constraint in Appendix Table 6 and Table 7. Under this evaluation, the pipeline achieves more reasonable success rates of 1.26% and 3.12% on the seen and unseen test sets, respectively, confirming our hypothesis. In this setting, InstructPro-1B still achieves the highest success rates, with further gains from scaling to 3B.

**InstructPro is scalable.** Scaling our model from 1B to 3B parameters leads to consistent performance improvements, with the overall success rate increasing from 2.46% to 5.06% on seen ligands and from 3.14% to 3.93% on unseen ligands. The larger model also achieves higher scores across most evaluation metrics, confirming that InstructPro follows the scaling law — demonstrating improved performance as model capacity increases.

## 5 ANALYSIS: DIVING DEEP INTO INSTRUCTPRO

In this section, we provide a comprehensive analysis to demonstrate the effectiveness of InstructPro, based on experiments conducted with InstructPro-1B on the unseen ligand test set.

### 5.1 DO BOTH TEXT AND LIGAND BOOST INSTRUCTPRO PERFORMANCE?

To assess the contribution of function descriptions and ligand SMILES to InstructPro's performance, we conduct an ablation study by removing the text encoder (InstructPro-w/o text encoder) and the ligand encoder (InstructPro-w/o ligand encoder). Results on the unseen test set are shown in Figure 2(a). Removing either component degrades both protein function scores (% < 2Å RMSD)

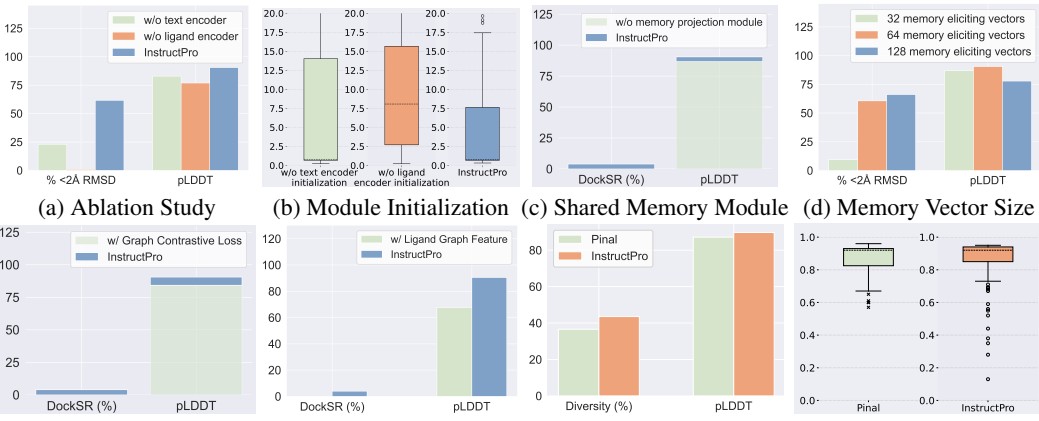

Figure 2: Ablation study on: (a) impact of text encoder and ligand encoder, (b) effect of using pretrained text encoder and ligand encoder initialization, (c) influence of shared memory module, (d) impact of memory eliciting vector number, (e) using additional graph contrastive loss during training, (f) using additional graph feature, (g) diversity and pLDDT of applying sampling strategy, (h) ipTM score of applying sampling strategy. **All experiments are conducted with InstructPro-1B on the unseen test set.**

and structural stability scores (pLDDT), with the absence of the ligand encoder causing a greater decline. These findings highlight that **both function descriptions and ligand SMILES are critical for designing ligand-binding proteins.**

To assess the impact of pretrained initialization for the text encoder and ligand encoder, we conduct an additional ablation study by training two model variants: one without text encoder initialization (InstructPro-w/o text encoder initialization) and the other without ligand encoder initialization (InstructPro-w/o ligand encoder initialization). The RMSD distributions of these variants are shown in Figure 2(b). The results indicate that removing either pretrained initialization leads to degraded RMSD performance, with the absence of ligand encoder initialization having a more substantial effect. These findings highlight that **both pretrained encoder initializations are useful for designing proteins that better align with the desired function descriptions**.

## 5.2 ROLE OF SHARED MEMORY MODULE IN ENHANCING INSTRUCTPRO

To assess the effectiveness of the shared memory module, we train a variant of the model that removes this component and instead utilizes the full set of text semantic features. As shown in Figure 2 (c), incorporating the shared memory module leads to improved performance, with higher docking success rate and pLDDT score. These results demonstrate that the **shared memory module effectively extracts critical information from the textual features, thereby facilitating more functionally consistent protein design**.

To explore the effect of memory eliciting vector size on model performance, we train two variants with 32 and 128 memory vectors, respectively. As shown in Figure 2(d), reducing the size to 32 results in clear performance degradation. Increasing it to 128 slightly improves the functional score (% < 2Å RMSD) from 60.62% to 66.14%, but substantially decreases structural stability, with pLDDT dropping from 90.518 to 77.855. To balance function and structure stability, we finally adopt a memory eliciting vector size of 64.

## 5.3 DOES ADDITIONAL GRAPH FEATURE OF LIGAND HELP?

Small molecules can not only be represented by SMILES strings but also as three-dimensional (3D) molecular graphs. To investigate whether incorporating ligand graph features enhances protein design, we first compute 3D graph representations of ligands using EGNN (Satorras et al., 2021), and then introduce a contrastive loss during training to align ligand graph representations with protein sequence

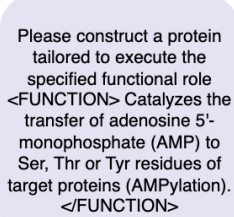 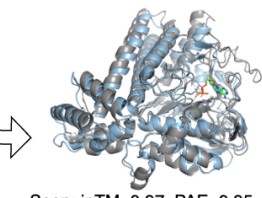 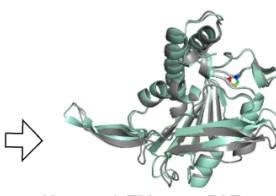

Figure 3: Ligand-binding proteins designed by InstructPro-1B. We omit ligand SMILES string in the figure due to limited space. In both examples, natural proteins are depicted in grey, while the proteins designed by InstructPro-1B are shown in distinct colors.

semantics. However, as shown in Figure 2(e), this additional contrastive supervision does not improve design quality. Next, we concatenate the ligand graph representations with the SMILES-based contextual features, and achieve decoding by conditioning on both the textual semantics and the combined ligand features. As illustrated in Figure 2(f), incorporating the additional graph features leads to both docking success rate and pLDDT score degradation. One possible explanation is that SMILES strings have already captured essential structural and bonding information of ligands, making the inclusion of explicit 3D graph features redundant or even conflicting. Thus, **leveraging SMILES alone appears sufficient for effective ligand-binding protein design in our framework**.

### 5.4 How Diverse Are the Designed Proteins?

To evaluate the diversity of proteins generated by InstructPro, we apply Nucleus Sampling (Holtzman et al., 2019) with a probability threshold of $p = 0.4$. For each function description and ligand SMILES, we generate five candidate sequences and compute the pairwise diversity score as $1-$ (amino acid recovery rate after pairwise alignment between candidates). As shown in Figure 2(g–h), InstructPro achieves a diversity score of 43.54%, surpassing Pinal's 36.59%. At the same time, proteins generated by InstructPro also achieve higher pLDDT values and ipTM scores. These findings indicate that **InstructPro can generate proteins that are both diverse and stable, while maintaining strong binding to the specified ligand**.

### 5.5 Case Study: Designing Ligand-Binding Proteins with InstructPro

Figure 3 showcases two ligand-binding proteins designed by InstructPro-1B, drawn from the seen and unseen ligand test sets, respectively. In both cases, the designs achieve ipTM scores above 0.9, PAE values below 5, pLDDT scores exceeding 90, RMSD values under 2, and high docking confidence, indicating near-perfect structural alignment with their natural counterparts. **These results highlight the capability of InstructPro to design ligand-binding proteins that accurately reflect human-specified functional intent.** We provide more designed cases in Appendix D.2.

## 6 Conclusion

In this paper, we introduce InstructPro, a natural language–instructable, multimodal framework for designing ligand-binding proteins. The framework consists of four core components: a text encoder, a ligand encoder, a shared memory module, and a protein decoder. Given a textual description of the desired function and a ligand formula in SMILES format, InstructPro generates protein sequences that are functionally aligned with the specified instructions and capable of binding to the target ligands. We develop and evaluate two variants, InstructPro-1B and InstructPro-3B, containing 1 billion and 3 billion parameters, respectively. Both consistently outperform strong baselines across multiple evaluation metrics. Notably, InstructPro-1B achieves the highest design success rates on both seen and unseen ligand test sets, while scaling to 3B yields further improvements. A limitation of this work is the lack of wet-lab validation, which remains an essential step for future research to confirm the model's practical utility.

ETHICS STATEMENT

We commit to releasing the curated InstructProBench upon acceptance of this paper to facilitate transparency, reproducibility, and further research in the community.

REPRODUCIBILITY STATEMENT

All code necessary to reproduce our results is available at `https://anonymous.4open.science/r/InstructPro-9B05`. In addition, the designed proteins generated by both InstructPro-1B and InstructPro-3B are included in the supplementary materials for reproducing our results.

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

# APPENDIX

## A  THE USE OF LARGE LANGUAGE MODELS

In this work, large language models are only used for typo correction and grammar checking.

## B  DATASET DETAILS

We provide detailed data statistics of our curated InstructProBench in Table 5.

Table 5: Detailed data statistics for InstructProBench.

| Split | Entry | Unique Uniprot ID | Unique Function Description | Unique Ligand |
|---|---|---|---|---|
| Training | 9,586,972 | 7,530,857 | 11,271 | 248 |
| Validation | 3,263 | 1,943 | 41 | 6 |
| Test (Seen Ligand) | 1,500 | 1,500 | 195 | 11 |
| Test (Unseen Ligand) | 127 | 127 | 43 | 5 |

## C  ADDITIONAL EXPERIMENTAL INFORMATION

### C.1  ADDITIONAL BASELINE IMPLEMENTATION DETAILS

We provide further implementation details of baseline methods as follows:

- **Pinal**: We just follow the instructions on their official GitHub repository to achieve design.

- **ESM3**: We begin by extracting functional annotations for each protein from InterPro (Blum et al., 2025). Using these annotations as conditioning signals, we apply ESM3 to generate an initial protein sequence and structure. Next, conditioned on both the functional annotation and the designed sequence, we refine the protein structure. Finally, leveraging the improved structure, we design a corresponding protein sequence. For evaluation, we employ the publicly available ESM3-open model, which contains 1.4 billion parameters.

- **RFdiffusion2+LigandMPNN**: Since there are no experimental protein-ligand complex structures for our seen and unseen test sets, we first apply AlphaFold3 to fold the complex structure for the target ligand and ground truth protein. Then taking the ligand structure from the complex as input, RFdiffusion2 designs a protein structure which binds to the given ligand. Applying the designed complex structure as input, LigandMPNN finally generates a protein sequence. Because RFdiffusion2 fails on ligand 1,2-diacyl-sn-glycero-3-phospho-(1'-sn-glycerol)(1-) in the unseen ligand test set, we are able to obtain only 96 designed proteins for this baseline.

- **AlphaFold3+LigandMPNN**: Similarly to RFdiffusion2+LigandMPNN, we first get the complex structure of the the target ligand and ground truth protein. Then taking the ligand structure and protein backbone structure from the folded complex as model input, LigandMPNN correspondingly designs a protein sequence.

- **ProGen2**: Since ProGen2 is an autoregressive protein language model without any conditioning signals, we provide the first 32 amino acids as a prefix to guide the model design.

### C.2  ADDITIONAL METRIC EVALUATION DETAILS

We provide further details about how we apply each metric to evaluate the designed proteins:

- **RMSD**: We first apply the ESMFold (Lin et al., 2023) to get the folded structure of the designed proteins and the ground truth, and then we utilize PyMOL (Schrödinger, LLC, 2015) to calculate their RMSD. Finally, the average RMSD on the whole test set is reported. Generally, if the designed protein has a RMSD < 2Å with the ground truth, it is highly likely the protein has the similar structure to it.

- **DockSR**: Similarly, we first predict the 3D structure of each designed protein using ESM-Fold Lin et al. (2023), and then perform docking to the target ligand using DiffDock. A docking score > 0 indicates a high-confidence docking.

- **AlphaFold3 ipTM** is an interfacial variant of the predicted TM-score, evaluating the interaction between different chains (Abramson et al., 2024). Values higher than 0.8 represent confident high-quality predictions.

- **AlphaFold3 PAE** estimates the error in the relative position and orientation between two tokens in the predicted structure. Higher values indicate higher predicted error and therefore lower confidence. A PAE score lower than 10 indicates a reliable structure.

- **AlphaFold3 pLDDT** aims to predict a modified LDDT score that only considers distances to polymers. It is a per-atom confidence estimate on a 0-100 scale where a higher value indicates higher structural stability.

# D  ADDITIONAL EXPERIMENTAL RESULTS

## D.1  SUCCESS RATE WITHOUT RMSD

Table 6: Performance on seen ligands. For limited space, RF2 denotes RFdiffusion2 and AF3 denotes AlphaFold3. **The results demonstrate that InstructPro-1B achieves higher success rate than baseline methods, and scaling the model to 3B yields additional performance gains.**

| Models | DockSR (↑) | ipTM (↑) | PAE (↓) | pLDDT (↑) | Overall Success Rate (↑) |
|---|---|---|---|---|---|
| ProGen2 | 2.06% | 0.589 | 10.496 | 71.655 | 0.26% |
| ESM3 | 2.66% | 0.624 | 10.181 | 75.368 | 0.00% |
| Pinal | 1.80% | 0.751 | 6.588 | 81.419 | 0.80% |
| RF2+LigandMPNN | 2.86% | 0.693 | 13.356 | 66.501 | 1.26% |
| AF3+LigandMPNN | 2.93% | 0.856 | 2.371 | 88.200 | 2.86% |
| InstructPro-1B | **3.13%** | **0.918** | **1.893** | **89.601** | **2.86%** |
| InstructPro-3B | **7.86%** | 0.877 | 2.261 | 87.586 | **5.60%** |

Table 7: Performance on unseen ligands (zero-shot setting). For limited space, RF2 denotes RFdiffusion2 and AF3 denotes AlphaFold3. **InstructPro-1B consistently outperforms all baseline methods on every metric. Moreover, scaling the model to 3B yields further performance improvements.**

| Models | DockSR (↑) | ipTM (↑) | PAE (↓) | pLDDT (↑) | Overall Success Rate (↑) |
|---|---|---|---|---|---|
| ProGen2 | 2.36% | 0.686 | 9.079 | 69.176 | 2.36% |
| ESM3 | 7.87% | 0.774 | 5.757 | 86.857 | 3.14% |
| Pinal | 0.00% | 0.814 | 3.247 | 87.157 | 0.00% |
| RF2+LigandMPNN | 4.16% | 0.745 | 9.898 | 75.491 | 3.12% |
| AF3+LigandMPNN | 3.93% | 0.868 | 2.923 | 89.862 | 0.00% |
| InstructPro-1B | **3.93%** | **0.869** | **2.638** | **90.518** | **3.14%** |
| InstructPro-3B | **3.93%** | **0.882** | **2.439** | 90.207 | **3.93%** |

The success rates without considering RMSD are reported in Table 6 and Table 7. When the RMSD constraint is removed from the overall success rate calculation, the RFdiffusion2+LigandMPNN pipeline achieves more reasonable scores of 1.26% and 3.12% on the seen and unseen ligand test sets, respectively. This supports our hypothesis that RFdiffusion2 is capable of designing novel structures that result in higher RMSD values, while still maintaining functional relevance and binding to the intended ligands. Nonetheless, InstructPro-1B continues to achieve the highest success rates, and scaling to the 3B variant yields comparable improvements.

## D.2  MORE DESIGNED CASES

Additional design examples are shown in Figure 4. All cases achieve ipTM scores above 0.8, PAE below 2, pLDDT above 90, RMSD below 2, and high docking confidence, highlighting InstructPro's ability to generate ligand-binding proteins that reliably align with the specified function descriptions.

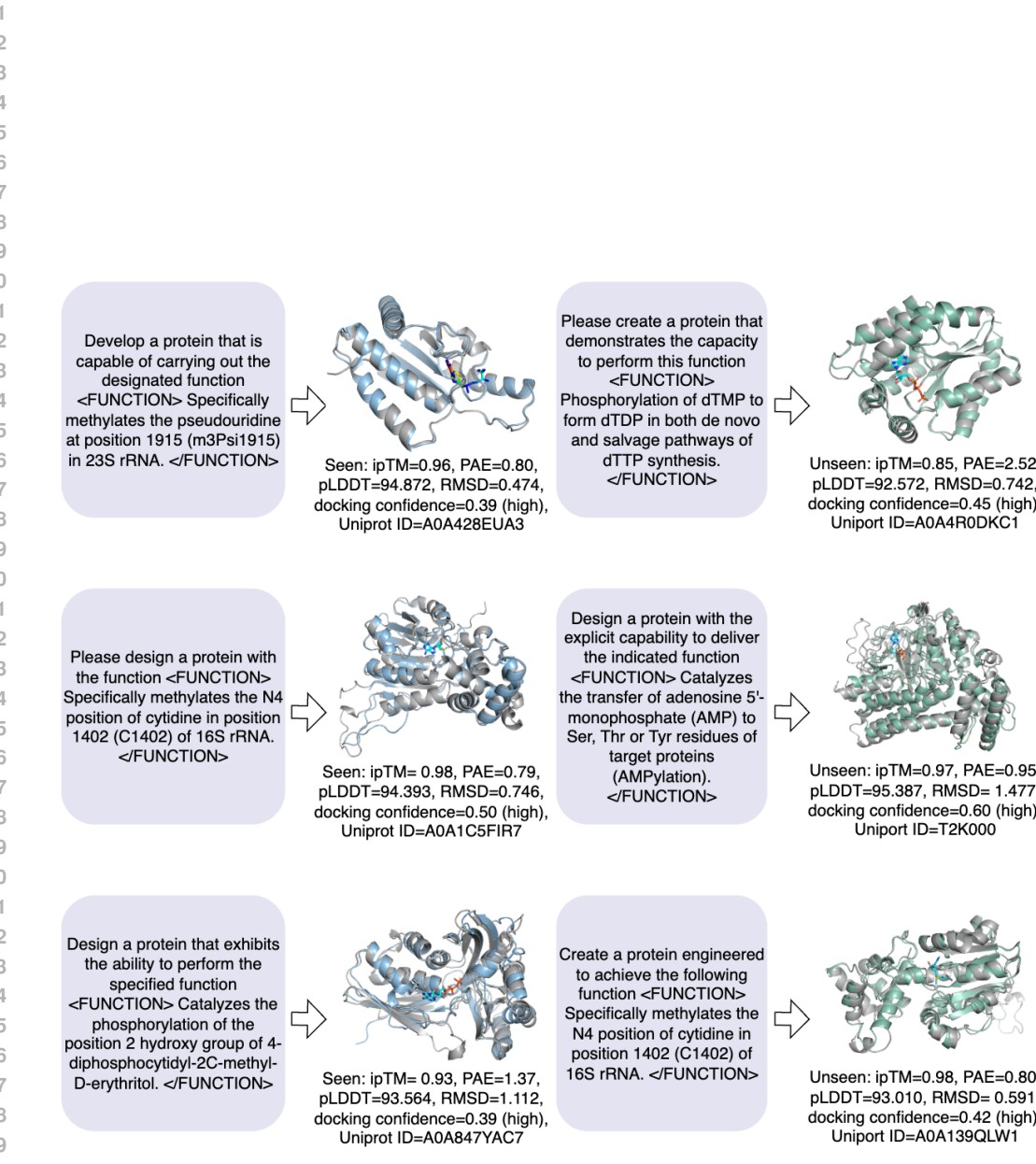

Figure 4: Ligand-binding proteins designed by InstructPro-1B. We omit ligand SMILES string in the figure due to limited space. In both examples, natural proteins are depicted in grey, while the proteins designed by InstructPro-1B are shown in distinct colors.

