# OpenReview forum: "InstructPro: Natural Language Guided Ligand-Binding Protein Design"
_ICLR.cc/2026/Conference — ICLR 2026 Conference Withdrawn Submission_

### Official Review · Reviewer_LhWG · 2025-10-29

**Soundness:** 1
**Presentation:** 2
**Contribution:** 2
**Rating:** 2
**Confidence:** 3

**Summary:**

This paper addresses the challenge of data scarcity in designing ligand-binding proteins by leveraging underutilized textual descriptions of protein-ligand interactions. The authors introduce **InstructPro**, a family of generative models capable of designing protein sequences from natural language instructions and specific ligand formulas.

To enable this approach, the paper presents **InstructProBench**, a large-scale dataset comprising 9.6 million triples of function descriptions, ligands, and proteins.

The authors trained two model variants, **InstructPro-1B** and **InstructPro-3B**, which substantially outperform strong baselines. The 3B parameter model achieves design success rates of 5.06% on seen ligands and 3.93% in zero-shot scenarios, demonstrating the potential of natural language-guided generative modeling to expand protein design capabilities beyond traditional data limitations.

**Strengths:**

1.  **Significant Contribution of a Large-Scale Dataset:** The development of **InstructProBench**, a large-scale dataset containing nearly 9.6 million (function description, ligand, protein) triples, is a substantial contribution to the field. By curating and releasing this resource, the authors are enabling further research and providing a valuable benchmark for the community to build upon.

2.  **Pioneering a Promising and Novel Research Direction:** The paper tackles the ambitious and interesting task of natural language-guided functional ligand-binding protein design. The proposed multimodal framework, which operates directly on natural language and ligand SMILES strings, presents a promising and innovative route for solving this problem. This approach could potentially bypass the limitations of structure-based design and opens up new possibilities for leveraging vast, unstructured biological text data.

**Weaknesses:**

### Weaknesses:

The primary weakness of this manuscript lies in its experimental design and evaluation protocol, which appears to be fundamentally misaligned with the stated goal of protein design. This raises significant concerns about the validity and interpretation of the presented results.

1.  **Fundamental Issues with the Evaluation Metrics:** The core metrics chosen—RMSD to a "ground-truth structure" and the proportion of designs with RMSD < 2Å—are appropriate for *structure prediction* tasks, not for *generative protein design*. The objective of protein design is typically to create novel proteins with desired functions, not to recapitulate existing ones.
    *   By penalizing structural deviation from a "ground-truth," the evaluation protocol incorrectly rewards memorization or retrieval of known structures rather than genuine, novel design. While tasks like inverse folding (e.g., LigandMPNN) also use RMSD, it is to verify that a designed sequence can refold to a *fixed input backbone*, which is a different goal from the task described here.
    *   This issue extends to the "SUCCESS RATE WITHOUT RMSD" metric in the appendix. Comparing a *de novo* generative model like RFdiffusion2 against a ground-truth database is not a fair comparison, as database retrieval is an inherently simpler task than generation. This metric conflates two different problems and unfairly penalizes models designed for true generation.

2.  **Inappropriate and Inconsistent Choice of Baselines:** The selection of baseline models is confusing and mixes models from entirely different task categories. The paper compares InstructPro against:
    *   **Sequence Generation Models:** ProGen2, ESM3
    *   **Backbone Generation Models:** RFdiffusion2
    *   **Structure Prediction Models:** AlphaFold3

    Comparing these models on a single set of metrics is scientifically unsound, as they are optimized for different objectives and solve fundamentally different problems. This "category error" in the experimental setup makes the reported performance improvements difficult to interpret and does not provide a meaningful benchmark of InstructPro's capabilities against relevant state-of-the-art design methods.

Taken together, these issues suggest a need to fundamentally rethink the experimental framework. The authors should reconsider what task their model is truly solving and design an evaluation protocol with appropriate metrics and baselines that can scientifically and effectively demonstrate its performance for that specific task.

**Questions:**

I do not have specific minor questions at this time. My primary concerns, which would require the authors' response, are detailed extensively in the "Weaknesses" section above. I believe the manuscript would be most significantly improved by addressing the fundamental issues raised concerning the experimental design, the choice of evaluation metrics, and the selection of baselines.

---

### Official Review · Reviewer_RNAN · 2025-10-29

**Soundness:** 3
**Presentation:** 3
**Contribution:** 2
**Rating:** 6
**Confidence:** 4

**Summary:**

The paper proposes InstructPro, a generative framework for designing proteins that bind to small molecules (ligands) based on natural language instructions. The model accepts a textual instruction and function description, along with a ligand formula expressed as a SMILES string, and generates a protein sequence that is intended to perform the specified function and bind the target ligand. To tackle the paucity of labelled protein–ligand complexes (~20 k structures), the authors construct InstructProBench, a large dataset of 9.6 M function–ligand–protein triples extracted from UniProt. They train two model variants:

- InstructPro‑1B, with ≈964 M parameters
- InstructPro‑3B, with ≈2.98 B parameters

The architecture contains four components: a text encoder that processes the concatenated human instruction and function description, a ligand encoder that embeds the SMILES string, a shared memory module that extracts salient semantic features from the textual embeddings, and an autoregressive protein decoder that outputs the amino‑acid sequence conditioned on the text and ligand representations.

**Strengths:**

- The work proposes a rather explored paradigm: designing ligand‑binding proteins directly from natural language instructions and ligand formulas. Integrating text semantics with ligand representations is an original contribution that extends prior language‑guided protein design, which mostly ignores small‑molecule conditioning.

- The authors compile InstructProBench, containing 9.59 M triples extracted from UniProt. They perform hierarchical clustering with MMSeqs2 to create non‑redundant training, validation and test splits. This dataset could become a valuable resource for the community.

- The combination of a Transformer‑based text encoder, a Transformer ligand encoder, a trainable shared memory module to extract salient textual features, and a large autoregressive protein decoder is well‑justified. The shared memory module reduces long‑text inefficiency by projecting the text into a small set of memory vectors, which significantly improves docking success and pLDDT scores.

- The paper evaluates across multiple metrics (RMSD, docking confidence, ipTM, PAE, pLDDT) and reports both seen and zero‑shot performance. Ablation studies test the role of each component (text encoder, ligand encoder, initialization, shared memory, memory vector size), examine adding 3D graph features, and assess diversity via nucleus sampling. These experiments provide insights into which design choices matter.

- Comparing 1 B and 3 B parameter models shows that increasing capacity improves success rates (2.46 → 5.06 % for seen ligands; 3.14 → 3.93 % for unseen ligands). This suggests that the method might benefit from scaling.

**Weaknesses:**

- **Low absolute success rates.** Even though InstructPro outperforms baselines, the overall success rates remain low: 2.46 % on seen ligands and 3.14 % on unseen ligands for the 1 B model. The 3 B model improves to 5.06 % and 3.93 %, but these values mean that more than 94 % of generated sequences fail to satisfy the full set of structural and functional criteria. As such, the practical usefulness of the model for generating viable candidates is questionable.

- **Reliance on predicted metrics and lack of experimental validation.** The evaluation pipeline depends on AlphaFold3 predictions for ipTM, PAE and pLDDT, and DiffDock for docking confidence. While these tools are state‑of‑the‑art, they remain in silico and are not perfect. An analysis of AI‑driven protein design notes that designs introducing novel folds or functions may have unintended biological effects and therefore require rigorous validation before use. The authors acknowledge that their work lacks wet‑lab validation, which limits the confidence in the designed proteins’ functional efficacy for practical cases (one of the main reasons for clinical failure).

- **Dataset bias and quality concerns.** InstructProBench is derived from UniProt annotations, which may be noisy and incomplete. Recent work [1] on protein language models shows that pLM likelihoods are biased towards species that are over‑represented in training data, causing designs to gravitate toward certain species and degrading properties like thermostability. The authors do not discuss potential species or ligand biases in their dataset, nor do they provide statistics on functional annotation quality. Without careful curation, training on a large but biased dataset may impair generalization to under‑represented proteins and ligands.

- **Limited zero‑shot evaluation.** The unseen‑ligand test set contains only 127 examples. Such a small sample may not adequately assess generalization. Moreover, the unseen ligands are drawn from the same protein families (clustered at 30 % identity) as the training set, which may not reflect truly novel chemistry. Evaluating on a broader range of ligands (especially those absent from training species) would strengthen claims of generalization.

- **Simplistic ligand representation.** The model encodes ligands solely via SMILES strings. The paper tries adding 3D graph embeddings and a contrastive loss but finds no benefit; however, this may result from the specific implementation rather than a fundamental lack of utility. Other studies have shown that 3D orientation and conformational flexibility are critical for binding affinity, and SMILES alone can miss stereochemical or tautomer information. Thus, concluding that SMILES are sufficient may be premature.

- *Absence of negative examples and explicit binding sites.* The generative model is trained only on positive triples (function‑ligand–protein). It does not learn what doesn’t work. Without negative samples or explicit binding site annotations, the model may memorise common motifs rather than learn causal relationships.


[1] Protein language models are biased by unequal sequence sampling across the tree of life

**Questions:**

- How does the model handle ambiguous or conflicting instructions? Natural language instructions may include ambiguous verbs or conflicting functional requirements. The paper does not provide examples where instructions are contradictory or require multi‑step catalysis. How robust is the model to poorly phrased queries?

- What is the distribution of ligands in InstructProBench? Providing statistics on ligand types (e.g., small molecules vs. ions), sizes, and binding mode diversity would help assess whether the training data supports generalization. Does the model perform equally well on hydrophobic and polar ligands? Could biases in ligand distribution explain the relatively low unseen‑ligand success rate?

- Are there species or taxonomic biases? The species‑bias study shows that protein language models trained on UniProt data favour sequences from over‑represented species, which can hurt design tasks. Did the authors analyse how the model’s performance varies across taxonomic groups? Might designs inadvertently drift toward mammalian sequences even when a microbial protein is desired?

- Why is the success rate so low even for seen ligands? A 2–5 % overall success rate implies that most generated proteins fail to meet structural or functional criteria. Which criteria are most often violated? Could alternative decoding strategies (e.g., beam search, nucleus sampling) or conditional filtering improve yield? A more detailed error analysis would be valuable.

- How sensitive are results to the choice of evaluation metrics? The authors define success as meeting thresholds on RMSD, docking score, ipTM, PAE and pLDDT. However, structural metrics such as RMSD may not correlate perfectly with function, and docking predictions can be noisy. How do success rates change if RMSD or docking thresholds are relaxed or replaced with other metrics (e.g., binding free energy predictions)? Appendix Table 6 shows that removing the RMSD constraint increases the RFdiffusion2+LigandMPNN pipeline’s success rate to 3.12 %, narrowing the gap with InstructPro. Similar sensitivity analyses for other thresholds would clarify the robustness of the conclusions.

- The model uses greedy decoding during inference. Did the authors compare against sampling or beam search? Given the extremely low success rates, exploring different decoding strategies might trade off between diversity and correctness.

**Suggestions for Improvement**

- Addressing dataset bias is critical. Future versions of InstructProBench could balance species representation and include curated annotations of binding sites, functional motifs and negative examples. Incorporating structural diversity from metagenomic sequences and synthetic constructs would reduce bias and improve generalization. The species‑bias study suggests that careful curation is necessary for meaningful design.

- Instead of treating SMILES and 3D graph features separately, consider co‑embedding them through graph neural networks or diffusion models that jointly model ligand and protein representations.

- Explore alternative decoding methods such as beam search, nucleus sampling or reinforcement‑learning‑guided decoding to increase the yield of viable candidates. Post‑processing with sequence optimization or iterative docking could refine the designs.

- Collaborate with experimentalists to test a subset of generated proteins. Even small‑scale validation would strengthen the claims significantly and provide feedback to improve the model, as emphasized by AI‑driven de novo design literature a lot.


InstructPro represents an innovative attempt to merge natural‑language instructions with ligand information for protein design. The architecture is thoughtfully constructed, and the authors carry out extensive experiments. However, the low success rates, reliance on computational evaluation, potential dataset biases and lack of experimental validation reduce the impact of the work.

Given these factors, I rate the submission 6 / 10. The paper is interesting and potentially useful, but improvements in dataset curation, evaluation rigor and experimental validation are needed before it can fully realise its promise.

If the authors address the weaknesses outlined above, or provide a right justification for the points raised in my questions, or incorporate any of the suggested future directions or clarify points I may have misunderstood, I would be willing to raise my score to an 8.

**Details Of Ethics Concerns:**

**Privacy, security and safety.** This work enables natural-language to protein generation for ligand-binding and plans to release code, data, and designed sequences without described access controls or hazard screening. Because the same pipeline could target hazardous ligands (toxins, delivery enhancers), and results rely on in-silico metrics without wet-lab validation, public release lowers the barrier to dual-use.

**Potentially harmful insights/methodologies/applications.** The paper provides actionable, step-by-step details (preprocessing, model specs, training, success thresholds) and demonstrates zero-shot design and diversity-boosted sampling, making replication easy for non-experts. Absent explicit guardrails, these methods could be repurposed to design binders or delivery proteins for sensitive targets.

---

### Official Review · Reviewer_RDGB · 2025-10-30

**Soundness:** 2
**Presentation:** 3
**Contribution:** 2
**Rating:** 4
**Confidence:** 4

**Summary:**

This work focuses on an important task, ligand-binding protein design. The authors propose InstructPro, leveraging the textual description from UniprotKB to facilitate the protein design. Additionally, to train and evaluate the model, the authors build InstructProBench, incorporating functional description, ligand, protein triplets. Experiments indicate InstructPro can achieve higher success rate compared with baselines.

**Strengths:**

1. The paper is easy to read and follow and the presentation is clear.

2. The authors build a pratical benchmark, which is beneficial for the following researches.

3. The experiments indicates the improved performance compared with baselines.

**Weaknesses:**

1. The success rates of InstructPro are still relatively low. Could the authors also introduce other baselines that are not based on deep learning?

2. The technical novelty is limited. InstructPro essentially combines the exsiting modules, such as PubMedbert, Roberta, and ProGen2, for a new task. Although the task is meaningful, InstructPro does not consider the inductive bias from such task.

3. The authors have tried to incorporate 3D structures by introducing EGNN but resulting worse performance. However, the task, ligand-binding protein design, should be a task based on 3D structure. Is that possible that, InstructPro actually does not understand the 3D structure in the protein and ligand?

4. The definition of SR metric seems problematic. The successful samples have to meet RMSD < 2A, but this criteria looks not that necessary. As the task is de novo design, such criteria may miss the novel design but with successful function.

**Questions:**

See Weaknesses.

---

### Official Review · Reviewer_FXjt · 2025-10-31

**Soundness:** 2
**Presentation:** 1
**Contribution:** 2
**Rating:** 2
**Confidence:** 4

**Summary:**

This paper presents InstructPro, a generative AI framework for designing ligand-binding proteins using natural language instructions and ligand formulas. The authors address the limitation of scarce protein-ligand structural data by creating InstructProBench, a large-scale dataset of 9.6 million text-ligand-protein triples. They introduce two models, InstructPro-1B and InstructPro-3B, that significantly outperform baselines. Notably, InstructPro-3B achieves design success rates of 5.06% (seen ligands) and 3.93% (zero-shot), demonstrating a promising new direction for protein design that leverages textual information.

**Strengths:**

- This work introduces a highly intuitive method that directly translates natural language instructions into functional protein sequences.
- The method's effectiveness is demonstrated with state-of-the-art results on a new, large-scale benchmark dataset, which itself is a major contribution to the field.

**Weaknesses:**

- The experimental results are not fully convincing. It is unclear why the evaluation computes similarity between the designed protein and the ground-truth structure.
- The performance of RFdiffusion2 as a baseline is surprisingly poor, despite being a strong model validated by wet-lab experiments.
- The diversity and novelty of the designed proteins are not reported, which limits understanding of the model’s generative capabilities.

**Questions:**

Q1 How do the authors assess whether the designed proteins truly satisfy the **natural language input constraints**? The provided examples involve strong structural and functional conditions, but the proposed evaluation metrics do not seem to capture these aspects.

---

### Note · Authors · 2026-01-23

I have read and agree with the venue's withdrawal policy on behalf of myself and my co-authors.